# Ableism, Human Rights, and the COVID-19 Pandemic: Healthcare-Related Barriers Experienced by Deaf People in Aotearoa New Zealand

**DOI:** 10.3390/ijerph192417007

**Published:** 2022-12-18

**Authors:** Michael Roguski, Tara N. Officer, Solmaz Nazari Orakani, Gretchen Good, Daniela Händler-Schuster, Karen McBride-Henry

**Affiliations:** 1Kaitiaki Research and Evaluation, Wellington 6012, New Zealand; 2Te Kura Tapuhi Hauora—School of Nursing, Midwifery, and Health Practice, Wellington Faculty of Health, Te Herenga Waka—Victoria University of Wellington, Wellington 6140, New Zealand; 3School of Health Sciences, Massey University, Palmerston North 4442, New Zealand; 4Institute of Nursing, School of Health Sciences, Zurich University of Applied Sciences, 8400 Winterthur, Switzerland; 5Institute of Nursing, Department of Nursing Science and Gerontology, Private University of Health Sciences Medical Informatics and Technology, 6060 Hall in Tyrol, Austria

**Keywords:** Deaf, disability, CRDP, COVID-19, Aotearoa New Zealand, healthcare, access, barriers, qualitative, human rights

## Abstract

The COVID-19 pandemic significantly affected global healthcare access and exacerbated pre-pandemic structural barriers. Literature on disabled people’s experiences accessing healthcare is limited, with even less framing healthcare access as a human rights issue. This study documents and critically analyses Deaf people’s healthcare access experiences in Aotearoa New Zealand during the COVID-19 pandemic. Eleven self-identified Deaf individuals participated in semi-structured videoconferencing interviews. Discourse analysis was applied to participant narratives with discourses juxtaposed against a human rights analysis. Barriers influencing healthcare access included: (1) the inability of healthcare providers to communicate appropriately, including a rigid adherence to face mask use; (2) cultural insensitivity and limited awareness of Deaf people’s unique needs; and (3) the impact of ableist assumptions and healthcare delaying care. Barriers to healthcare access represent consecutive breaches of rights guaranteed under the United Nations Convention on the Rights of Persons with Disabilities (CRPD). Such breaches delay appropriate healthcare access and risk creating future compounding effects. Action is required to address identified breaches: (1) The CRPD should also underpin all health policy and practice development, inclusive of pandemic and disaster management responsiveness. (2) Health professionals and support staff should be trained, and demonstrate competency, in Deaf cultural awareness and sensitivity.

## 1. Introduction

The COVID-19 pandemic has had a significant negative impact globally on healthcare access and has exacerbated pre-pandemic structural barriers, such as socio-economic disparities [1]. Such disparities have particularly impacted vulnerable populations, inclusive of disabled people [1,2].

Internationally, researchers have highlighted the unique pandemic experiences of disabled people, including significant disparities in delayed and unmet medical care [3,4,5], a situation that has been exacerbated by some health providers’ lack of knowledge about the unique needs of disabled people [4,6,7]. These experiences confirm the deleterious impact of disabled peoples’ marginalised status and need to be appreciated within the sombre reality that barriers to disabled people accessing healthcare has been associated with increased COVID-19-related mortality. For example, data from the United Kingdom demonstrate that disabled people account for 60% of all COVID-19-related deaths and the age-standardised mortality rate was significantly higher for disabled people than the general population [8].

Lessons derived from such inequities have centred on the urgent need to ensure that public health strategies are founded on person-centred responses, inclusive of the unique needs and experiences of disabled people [4,6,9,10]. Discursively, however, such recommendations are positioned within an apologetic frame and not as an inalienable right. As such, human rights are conspicuously absent from analysis of disabled peoples’ COVID-19 healthcare access experiences [6].

Notably, few studies have referenced human rights [6]. These studies, however, generally fall within a rubric of commentary and not primary research. For example, United Nations Convention on the Rights of Persons with Disabilities (CRPD) commentary, in association with nation-specific legislation and codes of practice, has been offered from Aotearoa (New Zealand) [11], South Africa [12], UK [13], and the USA [7]. Within this context, there is a need to explore disabled people’s experiences of accessing healthcare during the pandemic and juxtapose these experiences against a human rights framework.

Multiple rationales underpin this need. First, 1.1 million people in Aotearoa (24% of the total population) identify as disabled or as having an impairment [14], representing the nation’s largest minority group [11]. Despite access to care having been enshrined in a host of policy and legislative arrangements [15,16] disabled people have experienced inequitable access to health and disability support services [11]. Next, rather than an apologetic orientation, we argue that an adherence to rights is enshrined in the CRPD, and especially in relation to Article 25 that states that disabled people “have the right to the enjoyment of the highest attainable standard of health without discrimination on the basis of disability” [17]. Finally, on Wednesday 25 March 2020 at 11:59 pm, Aotearoa went into government-mandated isolation, employing a strategy of home “bubbles” [18] to limit physical contact within the immediate household for a two-month period and again in August 2021 for three weeks. Throughout this period, the government employed COVID-19 messaging, including specific messaging for the disabled community (see for example: https://covid19.govt.nz/prepare-and-stay-safe/iwi-and-communities/information-for-disabled-people/, accessed 13 December 2022 [16]), to convey a cohesive nation. It has been argued that the government’s repeated reference to ‘unity’, ‘kindness’, and the “team of 5 million” is a discourse reflective of embattled yet cohesive support [19,20]. However, Good et al. reflect that disabled people were generally forgotten, an invisibility that reinforces disabled peoples’ historical othering [21]; an invisibility that complements international accounts of a lack of government support [22] and unmet health needs compounded by poor public health messaging, lack of support services, and treatment postponement [13].

### 1.1. Deaf Peoples’ Experiences Accessing Healthcare during the Pandemic

The current study is a component of a larger and ongoing study on disabled peoples’ pandemic experiences. This paper focuses on Deaf ) peoples’ experiences; according to the 2018 census, those with a hearing impairment comprise 380,000 or 9% of the Aotearoa population [14]. ‘Deaf’ is used to denote people who identify as culturally Deaf and are actively engaged with the Deaf community. ‘Deaf’ is inclusive of a cultural identity amongst those with hearing loss who share a common culture, inclusive of shared understandings and meanings, and often includes a shared sign language [23].

A dearth of literature has been dedicated to Deaf experiences with COVID-19 and even less has focused on Deaf people’s experiences with COVID-19 within the area of healthcare access. Rather, health-related studies have centred on the inadequacy of public health communication and COVID-19 messaging and vaccinations [12,20,24,25,26]. At the heart of this inadequacy was the inconsistent or, in some cases, the absence of sign language and closed captioning [7,12,25,26]. Within this context, Dai and Hu (2022) documented the significant role of civil society in addressing a vacuum of public health messaging in China.

In terms of accessing primary and emergency healthcare, a number of barriers to Deaf people understanding healthcare provider communication have been identified. Profound communication barriers arose from infection minimisation policies that excluded family members and interpreters [12,26]; the impossibility of understanding healthcare workers wearing surgical masks [7,12]; and the assumption that telehealth is adequate or sufficient for Deaf people [7].

### 1.2. Aim

This study aimed to critically analyse Deaf people’s experiences in Aotearoa of accessing and engaging with healthcare during the COVID-19 pandemic between 2020–2022.

## 2. Materials and Methods

An in-depth qualitative research design was employed that was guided by a dual adherence to a social model of disability and an empowerment methodology to explore the lived realities of the Deaf, inclusive of the parents of Deaf children, in accessing healthcare during the pandemic. We defer to a social model of disability to highlight the role of ableism and the structural imposition of disability; namely imposed isolation and exclusion from full participation in society [13,27]. Empowerment methodology is used to acknowledge the socio-political history of disabled people’s marginalisation and stands in direct opposition to deficit models by reframing inquiry from deficits to competence, illness to wellness, and weaknesses to strengths. Similar to the social model of disability, empowerment research opposes models of individualised deficit or blame in favour of an appreciation of environmental influences. Empowerment methodology explicitly positions participant voices as central, regarded as valid and reliable [28,29].

### 2.1. Recruitment

Ethics approval for this study was granted by Te Herenga Waka—Victoria University of Wellington Human Ethics Committee (approval number: 0000030121). Participants were recruited using a purposive sampling and, thereafter, snowballing methodology. This entailed one of our research team (S.N.O.) liaising with a broad range of disability organisations and posting social media invitations. In addition, the remaining team members shared the invitation to participate through their networks. Eligibility criteria included: identifying as Deaf or being a parent of a Deaf child, being at least 18 years, and having had interactions with the health system during the pandemic.

### 2.2. Participants

Eleven people participated in the interviews, one of whom was a parent of a Deaf adult child with multiple complex needs. Participants’ ages ranged between 30 and 60 years. Most participants identified as female (n = 8), two as male, and one as non-binary.

### 2.3. Data Collection

Due to the ongoing presence of COVID-19, a combination of semi-structured individual and small group interviews (two interviews with three participants) were conducted online by S.N.O. over a three-month period in 2022, with the assistance of a New Zealand Sign Language interpreter.

The interviews followed a semi-structured format and centred on experiences accessing health care during the pandemic, experience of barriers accessing care, and the impact of those barriers on the individual’s health, as well as views on what needs to change to better help Deaf people access healthcare during a pandemic. The interviews ranged between 60 and 90 min and were audio recorded and transcribed with participant consent. Participants received a 50 NZD koha (gift) at the interview’s conclusion, in recognition of their time commitment. Transcripts were returned to participants, and they were invited to adjust the transcript as desired.

### 2.4. Data Analysis

A discursive approach underpinned transcript analysis. As described by Fairclough (2015), this process involved developing emergent themes focused on the presence and interplay of ubiquitous discourses [30]. The process involved attending to the participants’ context; focusing on the language participants used to describe their experiences; and centring on the emergent discursive concepts. Emerging discourses were consistently examined to determine the extent to which they were common across participants or differed according to participant-specific characteristics, including immigration status. In practice, this meant that discursive positions were created within an analysis framework. The analysis was initially conducted by M.R., refined further by G.G. and K.M-H.; after which the other research team members S.N.O., T.N.O., and D.H-S. reviewed and refined the analysis, drawing on their personal and professional clinical experience. Once these processes were completed, additional research team discussions further refined the analysis as some of the team drew on their lived experiences with hearing impairment. In keeping with the underpinning empowerment methodology, the analysis draws heavily on deidentified verbatim quotes that illustrate the emergent discourses. The research team adhered to the Consolidated Criteria for Reporting Qualitative Research (COREQ) quality assessment criteria, as created by Tong and colleagues [31] (see Appendix A).

## 3. Results

### 3.1. Pre-COVID-19 Discursive Context

Common across participants’ narratives were references to membership within the Deaf community and the community as a cultural entity, comprising unique needs, understandings, and language. Embedded within this cultural discourse were references to an embattled relationship with ableist norms and associated practices. 

Multiple accounts were shared whereby health professionals and support staff, inclusive of receptionists and security, have historically refused to accommodate alternative forms of communication. Notably, such refusal is counter to the CRPD definition of reasonable accommodation “necessary and appropriate modification and adjustments not imposing a disproportionate or undue burden, where needed in a particular case, to ensure to persons with disabilities the enjoyment or exercise on an equal basis with others of all human rights and fundamental freedoms”.

Rather, participants described pre-COVID healthcare experiences as privileging hearing forms of communication; a privileging that reflected a lack of cultural sensitivity, awareness of the unique needs of Deaf people, and a lack of priority afforded to the need for health and allied staff to learn even rudimentary sign language. Also common were accounts of being othered, whereby health and support staff failed to understand the significance or meaning of being Deaf and consequently engaged in demeaning behaviour.
It’s terrible. It’s treating the Deaf like they are stupid, and we are so not. That really gets me angry. They are like, “Caaan youuuu lip reeeead?” [talking slowly]. And it’s like, “Yes”. Come on, it’s our culture, guys, respect us. They’ve got no cultural sensitivity or awareness, they need to go off and learn that we are part of the community, we’ve got just as much right to be here as everybody else. They have to respect those that are in wheelchairs, they have to respect those that have other visual disabilities, deafness is not visual, and they just don’t respect us at all. (Immigrant, female, aged 50–59 #1)

Significantly, the embattled discourse was positioned in reference to ableism’s continued failure to adhere to the CRPD. In this sense, ableist beliefs, practices, and processes were described as infringing on Deaf peoples’ inherent human rights.
There’s the UN Convention that’s been signed, the CRPD and we should be following those sorts of things. That would make our lives so much easier and theirs.(NZ European male, aged 30–39 #1)

Furthermore, this specific discourse was imbued with active resistance against the imposition of ableist norms; denoted by participants demanding to be treated with respect and in accordance with their own cultural preferences. Examples were provided of people resisting these norms by choosing to leave a healthcare setting when they felt ableist practices and communication failed to accommodate their individual needs. In this sense, participants commonly confirmed the need for agency over receiving appropriate healthcare.
I’d like to have a regular doctor that I can easily communicate with. I used to have one when I was living in (region of Aotearoa). He was good. He actually respected me and talked to me because before that, growing up, I think as a baby that’s normal but I’m talking about probably 15 or 16. I’d go to the doctors, but my Mum and the old, old doctor would say to my Mum, “What’s wrong with you?” and not talk to me directly. “I’m here for my own issues. Don’t talk to my mother, talk to me. I can understand you”. We always told him, “Just talk to me directly”. He was like okay, “What’s wrong with you blah blah? Okay mother can you please ask her to...” What the hell? I just decided not to deal with them anymore and I found another doctor who actually just communicated with me directly. (NZ European, female, aged 30–39 #1)

The existence of Deaf culture and the pervasiveness of an embattled discourse framed participants’ pandemic-related healthcare experiences. However, within this contextual discourse, COVID-19 exacerbated existing ableist barriers associated with accessing primary, secondary, and tertiary care settings and other health professionals, such as dentists and occupational therapists.
I think the care and support was bad before the pandemic, the pandemic just made it worse. (NZ European, female, aged 50–59 #1)

The following analysis is structured according to four overarching discourses that emerged from the interviews, these are: (1) the inability of health providers to communicate appropriately, (2) cultural insensitivity and ableist assumptions, (3) the impact of ableist healthcare, and (4) access to healthcare as a human right.

### 3.2. The Inability of Health Providers to Communicate Appropriately

While participants understood mask use as a public health intervention, they framed masks as an ableist imposition that negatively affected communication with healthcare providers and associated health staff.
Visiting a GP [general practitioner] with the mask is frustrating. I understand they’re trying to protect themselves too, but they could choose those face shields and just have that. And then there would be better communication. And it’s their responsibility to have that. (Immigrant, female, aged 50–59 #2)

Within this context, masks profoundly impacted Deaf peoples’ understanding of what was said and resulted in heightened frustration, anxiety, and disempowerment.
I say, “I’m Deaf. Can you take your mask off so I can see what you’re doing? Can we write to each other?” They just ignore you keep the mask on, and you can see them talking through it. But you’ve got no idea what they’re saying. It’s like they don’t really get that we’re different. We cannot understand what you’re saying through a mask. (Immigrant, female, aged 50–59 #2)

Prior to COVID-19, most participants described being comfortable accessing healthcare services without an interpreter if the issue was not complex, critical, or overly confusing.
I would go for small issues. If COVID wasn’t there; for example, a toe infection or something like that. And it would be last minute GP appointments I could do that. But with my eyes and eye infection, or something like that, I would pop in to see a doctor because they wouldn’t have a mask on, so I could pick up some of the information on the lips. If it was very serious medical issue, yes… I definitely had an interpreter for those appointments.(Immigrant, female, aged 50–59 #1)

During the pandemic, because of infection control measures, the need for in-person interpreters increased; even for simple health-related interactions, with universal face mask requirements. However, the ability to access in-person interpreters during the pandemic became exceptionally difficult and mask use made communication exceedingly difficult, if not impossible for many. Participants were equally frustrated in situations when they could engage an in-person interpreter as interpreters also commonly wore masks; removing the ability to lip read and see facial expressions.
On one or two occasions we had a different interpreter who didn’t want to remove her mask, she was afraid of catching the virus and I really understand it. So I couldn’t communicate through her. (NZ European, female, aged 50–59 #1)

Participants also raised a preference for how their children preferred to be communicated with, as described in the following:
Yes, I would like to have the same interpreter, but it is not always possible. We do have a specific interpreter…, she’s the one that has a special mask [face shield] and my daughter’s comfortable with her and I am, and we would like to have her every time sometimes [name] is not available. That’s life. So, then I see who is available and contact the person that I’m most comfortable with. (NZ European, female, aged 50–59 #1)

#### 3.2.1. Non-Urgent Primary Care Narratives

Primary health practices reportedly differed according to the availability of interpreters, staff awareness, and understanding of Deaf peoples’ communication needs and preferences, and Deaf culture in general. Notably, such variability differed between provincial and urban settings, with a higher proportion of interpreters in urban settings. In the absence of interpreters, longstanding relationships with GPs or primary care providers led to providers and Deaf individuals having developed agreed modes of communication; these were premised on the provider’s generally understanding the unique needs of the Deaf individual. With the advent of COVID-19, an established relationship meant that providers made accommodations that minimised the individual’s stress and simultaneously empowered the Deaf individual.

Difficulties associated with accessing primary care arose when participants’ primary care provider failed to make reasonable accommodations, or the individual had to access medical providers with whom they did not have a pre-existing relationship; an issue that became exceedingly common throughout the pandemic due to staff shortages and increasing practice demands. In these situations, participants were unable to communicate because of the provider’s mask use and consequently, were prevented from accessing healthcare.
I had to go to my doctor’s appointment… their mask was on and I’m going, “Can you just take it off. I’m Deaf, I’ve got no idea what you’re saying. You’ve got the mask on; can you pull it down?” So, they did and then I could understand what was happening. The thing is, I had gone through four different people before I actually got to someone who was prepared to take their mask down so I could actually lip read. That was quite a shock for me, it wasn’t great. (Immigrant, female, aged 50–59 #2)

Importantly, accessibility was equally reliant on non-medical staff, such as receptionists and others gatekeeping entry into the building. In these situations, mask use and the failure to make reasonable accommodation resulted in heightened confusion and frustration, and in some instances, a failure to receive healthcare.

Notably, staff willingness to engage appropriately with Deaf individuals differed within practices. Such variability highlights the need for communication and cultural sensitivity training for all practice staff.
There are two nurses in the practice that I go; one is lovely, really soft and they do lots of gesturing, and the other one just wears a mask and doesn’t hear and she just wears the mask and won’t take it off and communicate with me. She treats me like a child, and it’s like, you know, I’m 50 I’m not a child. I know my health issues. I know what I need. I know what’s wrong with me. So, I always prefer the other one, I try to make sure that she’s there, and if she’s not there, I just go home.(Immigrant, female, aged 50–59 #2)

Alternative forms of engagement were also compromised by primary care providers who did not know how to use videoconferencing tools. Videoconferencing was the participants’ preference because conversing in a second language, such as written English, resulted in challenges.
I tried to email them, but the doctor was very ignorant about Zoom; didn’t really know how and said that they’d prefer to do it [consult] through email. (NZ European, female, aged 30–39 #2)

#### 3.2.2. Vaccination Narratives

Many participant discussions centred on their experiences with vaccinations and receiving booster vaccines, a process that required people to first select an available appointment time, register, and wait in a room with an unspecified number of others until their name was called, receive the vaccination, and then wait for a specified time so that adverse reactions could be monitored.

Heightened frustration, confusion, and in some cases panic, were associated with each stage of the vaccination process. Participants described how staffs’ use of face masks prevented them from understanding what was being said. Frustration was especially raised when vaccine staff repeatedly failed to adjust their form of communication to accommodate the individual’s request, a failure that resulted in negative mental health outcomes. Furthermore, many vaccination centres were reported as not having a process to book an interpreter.
When I went to get my vaccines and I had to write everything down. I said that I’m Deaf and explained if you want to communicate with me, please write it down or remove your mask. But they just ignored me. They just started talking to me through the mask and I had no idea what they were saying. It was so frustrating, and it really had a negative impact on my mental health and my wellbeing. It’s like they didn’t trust me. I’m Deaf and I need to know what’s going on. It was a really simple request, but they refused to do it.(Immigrant, female, aged 50–59 #2)

Significantly, considerable variation within practices was noted. Multiple participants described situations where staff failed to accommodate the individual’s communication needs and a sense of being “rescued” when another staff member intervened. The following narrative describes one participant’s experiences of two different vaccinations.
My first vaccination I went to one [vaccination centre]… I had a sign that said “I’M DEAF” but they [vaccine staff] didn’t really care; they just put it [the sign] to one side and carried on talking. One of them said, “Oh, come with me” and I got in first had my vaccine and I was out quickly so that was lovely, I felt treated quite well... The second vaccination that I had, they had the mask on the whole time and I had no idea what was happening and it was really hard to communicate with the staff there, I used lots of pen and paper. Today I went along to have my booster done and again I had information on my phone. They still kept their mask on and talked through it to me. I’m Deaf, I’ve got no idea what they’re saying so I pointed again to my phone, and they just indicated where to go. There was this massive queue and the lady said wait there in the queue, and slowly everyone got seen too. I had patience, but again, everyone was wearing masks, and nobody took any notice of my notice saying that I was Deaf, I got quite angry.(Immigrant, female, aged 50–59 #2)

Four participants described having at least one positive vaccination experience. Common across these experiences was an adherence to disability inclusive practices whereby staff acknowledged that the individual was Deaf and made necessary accommodations.
I’ve had the three vaccines. The first and third were positive. They took their masks down. They were very welcoming. They ushered me into where I needed to go, and I was able to tell them I’m Deaf. When I was ready to go they tapped me on the shoulder so I knew it was time to go. They actually had pen and paper organised for me telling me what steps I was going through at that time. The middle one wasn’t great. They refused to take their mask off and it felt quite cold. She just used her index finger and pointed where I had to go. When I was ready to leave, she still didn’t take her mask down. She talked to me but I had no idea what she said and so I left. But the other two experiences were very positive.(NZ European, female, aged 30–39 #2)

#### 3.2.3. Tertiary Healthcare Narratives

Similar to primary care and vaccination experiences, participants described extensive variability between and within secondary and tertiary care settings. Again, mask use and a failure or reluctance to accommodate alternative communication needs were a primary barrier.
When I went to the hospital to see the doctor three times, one experience was extremely positive. It was a great experience. The next two were extremely negative. They refused to take their masks down and we had to resort to pen and paper. It was really, really difficult. But the positive experience, it was at the hospital. They were extremely welcoming. They took their mask off. They made sure the interpreter was in place. They offered me coffee while I was waiting… That was a good experience.(NZ European, female, aged 30–39 #2)

In addition, those with multiple and complex needs who required tertiary health interventions struggled to communicate with various hospital departments during the pandemic. Historically, in-patient appointments had been the sole means of communication with specialists. With the advent of COVID-19, specialist appointments were greatly reduced or postponed indefinitely, which meant that participants were generally confused and greatly concerned about their health and ongoing treatment. Moreover, the inability to meet with specialists compromised the individual’s health and contributed to mental health challenges.
Just the lack of access, being able to get in and see people and talk to people. Because I mean, we all know that the hospital system is not the greatest at the best of times, they’re going through COVID. And you have to advocate for our kids. But advocating becomes very hard when you don’t have access to the actual doctors and the specialists. (Parent of NZ European, female, aged 18–24 #1)
I think mainly the problem with poor communication has caused me stress and anxiety. I had a problem with anxiety, and this [COVID-19] has made it worse.(NZ European, female, aged 50–59 #1)

In the case of a mother of an adult Deaf child with multiple complex needs, confusion, and the inability to communicate with varied tertiary health professionals was remedied by the parent identifying a senior hospital medical specialist who agreed to act as a conduit between various hospital departments.
The only real communication we’ve got was when we fought tooth and nail to get an overarching doctor and to have that access. So not everyone has that same access. But I can email him when I’m concerned or worried, and he normally gets back to me within 24 h, if not before.(Parent of NZ European, female, aged 18–24 #1)

In other situations, communication was aided when some interpreters used face shields, as opposed to masks, which greatly assisted communication. In other situations, some medical staff removed masks and engaged in social distancing.
Normally, if it’s my children’s consultations, I need an interpreter because that’s too confusing. Like doctors talking to my child is like a group situation, I can’t ask the doctor to talk to me and talk to my child at the same time. It’s much easier to have an interpreter. In those situations, I’ve asked the doctors to take the mask off, and the interpreter asks the doctor if it is okay to do the interpretation without the mask, so, we all take our masks off, but we keep a distance. And as soon as we are done talking, we put the masks back on again, and that works fine. (NZ European, female, aged 50–59 #1)

Finally, participants shared that there was a lack of post-procedural communication, assumed to be the result of increased workloads and efforts to contain viral spread. Nevertheless, participants described confusion, frustration, and feared not knowing their health status.
I have regular MRI scans and consultations with the neurology team. And it always used to be like they book your MRI scan, and after you have done the scan, you meet the team, and they explain the scan results… In the last two years, there has only been the MRI scans; no appointment or meeting afterwards. So, I went to my GP and asked, “Have you got a copy of my MRI results?” They said yes and printed a copy of the radiologist report for me. I read it and it said I had [a serious issue with] my spine, and it said a bunch of other stuff. It was really scary. (NZ European, female, aged 50–59 #1)

#### 3.2.4. Emergency Care Narratives

Most participants who required emergency care described negative experiences that resulted in heightened frustration, anxiety, and panic. These accounts differed from vaccination and primary and tertiary healthcare narratives in that the individuals were denied an interpreter because of infection minimisation policies. Consequently, and common across narratives, participants experienced treatment delays and were often in pain. Participants also shared accounts from across the Deaf community that reflected their collective experiences of emergency care.
One of my friends was rushed to hospital and she was in a lot of pain. She was in a lot of pain and they didn’t give her any pain relief for three hours while they were waiting for an interpreter. So, my friend was in a lot of pain for hours without the ability to communicate. She couldn’t communicate through written notes either.(NZ European, female, aged 30–39 #2)
We had an incident where [my daughter] ended up in A&E [Accident and Emergency] recently and because of the pandemic they wouldn’t let an interpreter in. I was like, “What the fuck?” It was very hard for her to find out what was going on for her care or why she was in the emergency. We know that you can’t have many people in the A&E but an interpreter is totally different.(Parent of NZ European, female, aged 18–24 #1)

### 3.3. Cultural Insensitivity and Ableist Assumptions

Participants generally described the pandemic as exceptionally stressful. While pandemic-related stress was acknowledged as affecting the general population, for Deaf participants, COVID-19 stresses made service engagement more anxiety-provoking than normal, which meant “everyday” tasks became extraordinarily difficult. Within the context of exacerbated stress levels, participants shared accounts of ableist assumptions that reflected an inflexible medical system, often founded on assumptions that failed to deliver services in accordance with individual needs, culture, and language. Inadvertently, these assumptions resulted in heightened stress, frustration, and reluctance to engage with healthcare.

Considerable attention was drawn to the assumptions surrounding language. The healthcare system was described as assuming Deaf people can readily engage in English, an assumption that failed to appreciate that Sign Language is a unique language denoted by its own grammatical and sentence structures and is often the individual’s first language. Within this context, and exacerbated by COVID-related stress, participants struggled to complete what ableist structures would assume to be rudimentary tasks.
English isn’t the strongest language for Deaf community.(NZ European, male, aged 30–39 #1)
Communication with the doctor is a barrier, with the nurse is a barrier, filling out the forms; sometimes I struggle with the forms. My English is good but sometimes I look at it and am just like, when I’m panicking or stressed, I just refer straight back to my native language, which is Sign Language. English is my second language as well. So sometimes when I’m panicking and I’m looking at this form it’s just a blur. I can’t do this.(NZ European, female, aged 30–39 #1)

In other situations, assumed language comparability placed Deaf people in precarious situations when they were pressured to translate. For example, one participant described taking their partner to a hospital emergency department. Even though an interpreter had been requested, concerns over their partner’s health meant they were asked to translate. While the participant understood that it was a potentially urgent situation, she felt compromised in her role as a support person and in her ability to adequately translate, a situation that potentially placed their partner at risk. Notably, such pressure to translate negatively impacted on the participant’s mental health and wellbeing.
There was one time when he [partner] had heart flutters, like palpitations. I hauled him off straight to A&E. When we arrived, we asked for an interpreter and of course it takes ages for the interpreter to turn up. They had to deal with this right now, so they used me to interpret—asking a Deaf person to relay information to another Deaf person! I’m going, “Oh god, I don’t know what to do”. That put a lot of stress on me which, I look back and just went, that’s not fair. I should be comforting or looking after my partner at the time, but I was kind of shot into this place of being the interpreter because he had to give his consent and information. It was quite hard for me… (a) I am not qualified for this you know; and (b), this is my partner that we’re talking about. Every time he gives an answer, I’m like, “No, no, you missed this”. So, I’m purposely changing his answer because that’s just what you do as a girlfriend. So, I found that quite difficult. I can understand from the medical point of view they needed to find a way to communicate with the Deaf person straight away. But at the same time, think about the support person. Sometimes the support person is actually not that supportive if you put a lot of pressure on them. (NZ European, female, aged 30–39 #1)

Significant commentary centred on interpreters and ableist assumptions associated with service provision. Similar to ableist assumptions surrounding supposed language comparability, participants stressed that it was erroneous to assume that all interpreters can provide medical-related translation, both in terms of the translator’s familiarity with medical terminology and their ability to translate presenting issues adequately to people who may be unfamiliar with the vocabulary.
Having qualified and appropriate interpreters is probably key. You can’t really just take a university grad interpreter and put her in the middle of a really heavy situation. You really can’t take a performance interpreter. You know, that’s really good on stage telling a story. Expect them to interpret medical. The same thing goes for court [interpreters]. We’ve got interpreters that specialise in court only. They come into the doctors, and they don’t know the medical terms. I know the interpreters and I know their skills and their experience. When I go an appointment, I get in contact with three people [interpreters] because I know they will be able to handle this appointment. I won’t contact the others because they are not relevant to this appointment. So, interpreters are a lifesaver in healthcare but they’ve just got to be used appropriately.(NZ European, female, aged 30–39 #1)

The importance of interpreters being sufficiently trained was further supported by the fact that the Deaf community differs in terms of language aptitude, ranging from “grassroots” to “academic”. In addition, older Deaf individuals were described as using older forms of sign language and, therefore, required different types of signing. Such differences were exacerbated by regional variations.
They’re older Deaf and they’ve used a different language as they’ve grown up. Times have changed and maybe their signing skills aren’t that great. Then we have a group that called Deaf Plus that have an added disability as well so being able to understand that jargon that’s happening, I wouldn’t think so. Older Deaf people need different types of signing.(Immigrant, female, aged 40–49 #1)

As a consequence, participants stressed wanting to choose their own interpreter; a requirement that can be understood in terms of the interpreter’s perceived aptitude, as an acknowledgement of past rapport, an appreciation of the way the interpreter interacts, and as a way to avoid disclosing personal, and potentially embarrassing, information to multiple people.
Organizing interpreters is sometimes difficult and it has become more difficult because I like to be able to choose my interpreter. Particularly when we need to work with my daughter, because she is very sensitive…. I’m not comfortable with those interpreters who I don’t know personally or socially. So, I want to be able to pick my interpreter, but it is not always easy. Especially at [local tertiary] hospital, there is a big push to use their own interpreters…When we have an appointment there, I tell the department that we do not need an interpreter… I will contact the interpreter I want and ask them to hold a time for our appointment. Then at the last minute and just before our appointment we tell the department that we need an interpreter, and obviously no one is available except for the interpreter we want. So, that is how I make sure we have the interpreter we want.(NZ European, female, aged 50–59 #1)

Participants also raised a number of service delivery assumptions associated with the provision of telehealth and videoconferencing, the assumption being that such provision would meet the needs of those who are Deaf and hearing impaired. Many participants shared a dislike of telehealth and audio-visual translation services because they prevented the individual from knowing what was being said about them. In these situations, participants described being relegated to the status of a passive third party. Instead, participants preferred an in-person translation service that ensured their active participation. In this sense, participant narratives reflected an ableist assumption that such mechanisms are sufficiently inclusive and endorsed by the Deaf. Effectively, such assumptions position Deaf persons as passive recipients of healthcare interactions, rather than active participants in their own health.
With the screen, things are sometimes okay. But the issue is, we can’t hear what’s being said. We keep saying, “Oh, we can’t hear what you’re saying”. “It’s too noisy”. “We can’t hear anything”. Yeah, the background noise. That’s why face-to-face is so much better for us. Or maybe you can’t see the screen properly. It’s pixeled out. There’s lots of issues. And it’s extremely stressful added to the reason for needing to go to the doctor as well.(Immigrant, female, aged 50–59 #2)

For some participants, telehealth or video conferencing were also disliked as they require additional time, which extends GP appointments. Consequently, participants described feeling rushed by the GP and uncomfortable, perceiving their provider to be frustrated trying to address their needs within an allocated timeframe. Significantly, participants found themselves in a paradoxical position of both appreciating and disliking telehealth, which meant some refused to use this modality.
And the GP is not prepared to wait, they want to quickly get through things.(NZ European, female, aged 40–49 #1)

Some participants described refusing to use telehealth or videoconferencing, with an offsite interpreter while they met with their GP, because of an associated financial burden. Participants’ narratives reflected an ableist assumption that the Deaf individual should bear financial responsibility for costs associated with accessing communication.
It’s using my personal mobile data, and that’s really expensive. A lot of GPs refuse to give us their Wi-Fi password. So, we’re expected to use our own data. That doesn’t make sense. (Immigrant, female, aged 50–59 #1)

Within a context of marginalised status, some participants shared extraordinary levels of vulnerability. Many participants brought hearing family members or support people to medical appointments to help navigate communication barriers; a situation that some participants described as unethical as it placed an unfair responsibility on these individuals. Such ethical dilemmas were especially noted when children were placed in a position of interpreter.
I just force my son to come and interpret for me. It’s not right. It’s not fair on him, but I just have to, so I’ll get my son [aged under five years] to interpret. He’s still really a child, but I get him to come and interpret for me in medical situations... it’s just because I couldn’t get another interpreter. (Immigrant, female, aged 50–59 #2)

However, those without family or a readily accessible support networks described intense isolation and being reliant on provider accommodation and the ability to access translation services. The most vulnerable cohort, however, were foreign nationals without permanent residency status, as this group was not entitled to access State-provided translation services. For this cohort, the cost of translation was prohibitive and represented a significant healthcare access barrier.
I don’t have permanent residency. You must be a resident before you get any support in New Zealand. There are no disability services that would support or help me at all. You know, hearing, they can get it. But I’m Deaf, no, they won’t give it to me. I have had to pay for interpreters myself. But I have been working here, paying tax, and I should have been allowed access to the [agency] funding.(Immigrant, female, aged 50–59 #2)

### 3.4. The Impact of Ableist Healthcare

Difficulties communicating because of mask use, a concomitant lack of accommodation, and profound difficulties accessing interpreters were described as insurmountable and often resulted in participants disengaging from health services during the pandemic.
I tried not to see doctors during COVID. It was too hard getting interpreters. I feel like I can’t communicate because of the attitude at the doorway. The doors [to the clinic] are closed, and they open them a little bit and they chat. And it’s like, “What?” “Can you pull your mask down so I can see what you are saying?” They won’t. They just won’t communicate with us. having no interpreter is worse, we don’t get any information.(NZ European, female, aged 40–49 #1)

Within this context, participants described only accessing healthcare in more critical situations. Such reticence was described as having important implications for the short and long-term health of the Deaf because it introduces additional disparities.

### 3.5. Access to Healthcare as a Human Right

Participants commonly drew on an embattled discourse to reflect their continued efforts to access care. Notably, an essential component of this discourse was the individual’s rights as juxtaposed against monolithic ableist structures. The centrality of human rights is evident in the various amelioration strategies suggested by participants. Such strategies address a lack of awareness and cultural insensitivity among health-related staff and the impact of communication barriers.

#### 3.5.1. Awareness and Cultural Sensitivity

Complementing CRPD [17], Article 8 (2d) (“Promoting awareness-training programmes regarding persons with disabilities and the rights of persons with disabilities”), participants requested that those who work in health settings, inclusive of receptionists and security staff, be trained in Deaf inclusive practices.
They need Deaf awareness training. That really should be part of their training. (NZ European, male, aged 50–59 #1)

In addition, as aligned with Article 25 (d) (“Require health professionals to provide care of the same quality to persons with disabilities as to others, including on the basis of free and informed consent by, inter alia, raising awareness of the human rights, dignity, autonomy and needs of persons with disabilities through training and the promulgation of ethical standards for public and private healthcare”), participants especially noted that awareness training should include cultural sensitivity and training on ableist assumptions. Special reference was made to language preferences: that NZ Sign Language is commonly a Deaf person’s first and primary language and that many Deaf people are not proficient in English. Participants also stressed the importance of interpreters, and the impact of anxiety negatively impacting on the individual’s ability to comprehend health professionals’ communication.
For Deaf people it can be really difficult to communicate in written English so that’s a barrier as well, even online. It’s really important doctors know that Deaf people need interpreters and some just don’t have that knowledge, they don’t have that awareness.(Immigrant, female, aged 40–49 #1)
It’s really important to have access to someone that can communicate clearly. There’s lots of emotions that are going on. We’ve got high anxiety in those areas too. Yeah, so communication is really important. And you know, we’ve got high anxiety when we’re going to the doctor in an emergency situation, so we need to calm down and be able to focus and actually communicate clearly. That’s why we need the interpreter there.(NZ European, male, aged 50–59 #1)

#### 3.5.2. Removal of Communication Barriers

Masks were universally discussed as a primary impediment to communication and were identified as a significant barrier to engaging with health-related staff. Complementing CRPD [17], Article 9 (e) (“To provide forms of live assistance and intermediaries, including guides, readers and professional sign language interpreters, to facilitate accessibility to buildings and other facilities open to the public”), participants strongly advocated for health and support staff to use large, non-reflective, face shields. They further suggested that interpreters be provided with large face shields.
Interpreters should have provided those shields. So, it’s really frustrating because some interpreters refuse to take their mask down as well. And it’s only related to health… they use face shields in most medical settings. Interpreters should be provided with them if it is a medical appointment.(Immigrant, female, aged 50–59 #2)

Also aligned with Article 9 (e), participants requested universal provision and increased access to in-person interpreters and, in accordance with Article 4 (h) (“To provide accessible information to persons with disabilities about mobility aids, devices and assistive technologies, including new technologies, as well as other forms of assistance, support services and facilities”), interpreters via audio-visual platforms. Notably, it is the individual’s choice whether they choose to engage with in-person or videoconferencing interpretation services. Also complementing Article 9 (e), participants stressed that such provision should not result in a financial burden to the individual, and should, therefore, be inclusive of all residing in Aotearoa and not reliant on immigration status.
It’s really important to have access to someone that can communicate clearly. There’s lots of emotions that are going on in those areas…They need to have interpreters on standby there for us just in case they’re needed. And often when you go to an emergency situation, we’re told there’s no interpreter. There’s no one there for us. So we have to wait for hours and hours and hours where we could have been seen and gone home quite quickly. It’s a high risk for us in the health area. It doesn’t matter if it’s on the weekends or night time, they should be based there.(NZ European, male, aged 50–59 #1)

Participants strongly advocated for the reversal of discriminatory and inequitable policies that prevented interpreters from supporting an individual while in hospital (Article 2 (“Discrimination on the basis of disability’ means any distinction, exclusion or restriction on the basis of disability which has the purpose or effect of impairing or nullifying the recognition, enjoyment or exercise, on an equal basis with others, of all human rights and fundamental freedoms in the political, economic, social, cultural, civil or any other field. It includes all forms of discrimination, including denial of reasonable accommodation”). This was especially raised when hospital policy excluded interpreters on the basis that such inclusion would be counter to pandemic restrictions. It was noted that such practices negated an individual’s dignity and autonomy while countering principles of inclusion, equality of opportunity, and accessibility (see Article 3).

The caveat to interpreter provision, however, was that interpreters should meet a specified standard of capability, inclusive of medical terminology, and the ability to communicate with others who have variable sign language proficiency.
I would like to see the proper review procedure for the current interpreters with the District Health Board, because I am not the only one that notices the substandard application of those interpreters.(NZ European, female, aged 30–39 #1)

Finally, complementing principles of inclusive practices (Article 3), participants advocated for inclusive communication practices in acknowledgement of Deaf individuals often compromised ability to readily access information.
Follow-ups and communication; making sure nothing is lost or overlooked when the system is under pressure. (NZ European, female, aged 50–59 #1)
The other thing that works really well is having, like people keep in communication with us. For example, this is we haven’t had but like, with the whole hip surgery, even just flicking us an email to let us know that we still haven’t got a date. We’re still you know, trying to work out what’s going to happen with the system or whatever is going on. (Parent of NZ European, female, aged 18–24 #1)

## 4. Discussion

COVID-19 exacerbated hegemonic ableist communication that made accessing healthcare inordinately difficult for the Deaf. Significantly, when viewed as a Gestalt of health access experiences, Deaf participants’ accounts highlight a series of human rights infractions that demonstrate a breach of rights guaranteed under CRPD. Further, rather than a cohesive “team of 5 million” [19,20], Deaf people are framed as other; a continuation of a historical legacy of invisibility and marginalisation [21].

A number of factors were identified as having an especially significant impact: the inability of health providers to communicate appropriately, including the rigid adherence to face mask use, cultural insensitivity, a variety of ableist assumptions, and the impact of ableist healthcare. Of concern is the ensuing delay in healthcare resulting from these structures and the risk of future compounding effects emanating from these delays. In general, researchers have highlighted similar challenges for disabled people in accessing healthcare [4,6,12,13,32,33,34]. Indeed, Saunders and colleagues’ survey demonstrated that face coverings negatively impacted healthcare-related communication [33]. However, this is the first study to use a qualitative design to focus solely on the experience of Deaf people accessing healthcare during the COVID-19 pandemic.

A high degree of variation was noted between and within practices, and also between regions. In this sense, some health-related staff were aware of the unique needs of the Deaf community and proactively engaged in Deaf inclusive forms of communication. What is disturbing, however, is that positive examples were an exception, and many accounts were shared where healthcare providers and support staff were unwilling or ignorant of the Deaf community’s communication needs which, in many cases resulted in participants experiencing extreme frustration, confusion, and panic. In other cases, the failure to appropriately communicate resulted in delayed and unmet medical care [5,35]. Equally disturbing were accounts of Deaf individuals being in pain for prolonged periods of time because of the failure of health service to provide appropriate accessible healthcare because they were refused accessible methods to support communication. Moreover, the inability to meet with specialists compromised the individual’s health and contributed to mental health challenges. Finally, a lack of post-procedural communication compromised wellbeing and created unnecessarily heightened stress.

As with all qualitative research, generalisability from this study is not possible, nor is it the aim. This study offers a globally unique insight into the challenges of accessing healthcare for the Deaf community during a pandemic. Such challenges are compounded by intersecting vulnerabilities experienced by marginalised populations [6,11].

Immediate action is required to address the identified human rights infractions. The CRPD should underpin the development of health policy and practice, inclusive of pandemic and disaster management responsiveness. The CRPD requires the inclusion of disabled people, inclusive of the Deaf community, in hospital and health service policy and planning around pandemic and disaster management (cf. [3,6]). Similarly, there is a need to reconsider how practices operate to make environments more accessible for Deaf people. Our participants highlighted the importance of existing relationships with trusted health providers, that interpreters should be included as an integral component of healthcare teams, and that barriers to communication, such as masks, would be alleviated with face shields. In addition, holistic service provision requires health providers to have information about a Deaf person’s preferred interpreters, primary care provider, and that the Deaf person has access to information in a time and manner that works for them.

The primacy of person-centred care and responses ensures the healthcare is centred around the unique needs and cultural preferences of disabled people [4,6,9,10,36]. Within this context, there is an urgent need for health professionals and support staff to be trained, and demonstrate competency, in Deaf cultural awareness and sensitivity. Further, education initiatives must focus on how to deliver services for people with less visible needs, but also to create environments that are more accessible for Deaf people.

## 5. Conclusions

While COVID-19 placed considerable stress on global health systems, the system’s response to the pandemic exacerbated existing inequities and created often inordinate barriers that made accessing healthcare exceptionally difficult for Deaf individuals. Lessons derived from the pandemic highlight the urgent need for immediate action to rectify considerable human rights infractions. There is also a need to increase the healthcare system’s awareness and cultural sensitivity of Deaf cultural needs and communication preferences and remove a raft of communication barriers. This can be achieved through targeted Deaf cultural awareness and sensitivity training, and by creating accessible environments for Deaf people. Failure to do so will result in the continued marginalisation of a significant proportion of the population and the continued denial of equitable health provision as guaranteed under the CRPD.

## Data Availability

Not applicable. The de-identified datasets used and/or analysed during the current study are available from the corresponding author on reasonable request.

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
