# Peer review of "Ableism, Human Rights, and the COVID-19 Pandemic: Healthcare-Related Barriers Experienced by Deaf People in Aotearoa New Zealand"

_ijerph, 2022, doi:10.3390/ijerph192417007_

Round 1

Reviewer 1 Report

The article submitted for review raises an important and difficult problem regarding the rights of people with disabilities, and in this case their access to medical care. The authors' great involvement in the research is visible, and their lack of acceptance of discrimination is evident, especially in the introduction of the article. However, it is important to make a few corrections that will allow a better understanding of the text.

The introduction should include information on the rights of people with disabilities in Aotearoa New Zealand resulting from national standards (outside the UN Convention). As well as information on how the medical care of people with disabilities (including the Deaf) functioned before the pandemic.

I propose to delete "documentation" for the purpose of research, a critical analysis will suffice.

Methodology. All abbreviations used in this chapter require explanation. The answer should also be supplemented with information on: How long did the recruitment for the research last?

How many individual interviews were there, and how many in a small group - small is how many people?

Did the same people participate in the additional team discussion?

Does the personal and professional experience of part of the research team result from their education, employment, are they people with hearing dysfunction?

I suggest changing the title of the Findings chapter to Research Results.

The division into subsections of the RESULTS chapter is incomprehensible, what does it result from? In the research methodology, in the topics of the interviews, but also in chapter 3.1. there is no word, for example, about COVID 19 vaccinations, and yet it functions as a separate subsection.

Discussion. It should try to analyze why, despite such a large population of Deaf people in Aotearoa New Zealand (380,000), only 11 people applied for the study.

Conclusions. Please add information on how to increase awareness and sensitivity of medical staff to the needs of Deaf people.

Author Response

Many thanks for your insights. Please see the attached table for our responses. 

Reviewer 2 Report

This article provides analysis from qualitative interviews with 11 Deaf people’s experiences accessing healthcare in Aotearoa NZ during the COVID-19 pandemic. An appropriate methodology of discourse analysis was used, within a human rights framework to interpret the data. The results showed: (1) the inability of healthcare providers to communicate appropriately to Deaf participants, (2) cultural insensitivity and limited awareness of Deaf people’s unique needs; and (3) the impact of ableist assumptions and delayed healthcare. These results outline a breach under the United Nations Convention on the Rights of Persons with Disabilities (CRPD).  Recommendations: 1) The CRPD needs to underpin all health policy and practice development, including during pandemic and disaster events. 2) Health staff should be trained, and demonstrate competency, in Deaf cultural awareness and sensitivity.

Review response required for the following points:

Introduction

1)      Paragraph 3: Lessons learned from inequities …are founded on person-centered responses. What evidence do the authors have for the statement ‘such recommendations are framed within an apologetic frame and not as an inalienable right.” ? Please clarify this further

2)      Paragraph 5: Covid-19 messaging,

a.      “Aotearoa went into government-mandated isolation, employing a strategy of home “bubbles” to limit physical contact within the immediate household…”- ‘bubbles’ needs explanation and referencing of the source of ‘bubble concept’

b.      government employed COVID-19 messaging- the authors have not included the governments disability-specific messaging (see Ministry of Health Covid website) which needs inclusion here and referencing

3)      As part of Ti Tiriti o Waitangi Tāngata Turi (Deaf Māori) are a critical part of Aotearoa New Zealand population but the authors have no mention in this introduction. Please explain?

Methods

Appropriate choice of methods has been employed with good explanation of the social model of disability and avoidance of deficit models.

4)      Participants: More details are need on demographics of participants recruited e.g. age, ethnicity, rural/urban residence etc

Analysis

5)      Analysis by ethnicity is not included- this needs to be included here.

Results

Overall, the results were presented clearly, and supported by relevant quotes. This included related analysis with articles from the UNCRPD. 

6)      There was no mention of any other culturally-specific issues for tāngata turi or other minority cultures. An intersectional lens needed to be included here

7)      Quote “That’s a bit of a clash for culturally Deaf signers. (Immigrant, female, aged 40-49 #1) needs contextual explanation to clarify the relevance.

Discussion

Relevant points that link to the data are discussed. There is a gap in research related to Deaf experiences during Covid-19 and future pandemic planning including Deaf communities.

8)      The authors generalize by their statement that “This study offers a globally unique insight into the challenges of accessing healthcare for the Deaf community in Aotearoa during a pandemic.” This needs further qualifying to the specific community referred to e.g.  ‘the non-indigenous Deaf community in Aotearoa’.

Author Response

(The authors gave the same response as above.)
